# Functioning of People with Lipoedema According to All Domains of the International Classification of Functioning, Disability and Health: A Scoping Review

**DOI:** 10.3390/ijerph20031989

**Published:** 2023-01-21

**Authors:** Lise Maren Kloosterman, Ad Hendrickx, Aldo Scafoglieri, Harriët Jager-Wittenaar, Rienk Dekker

**Affiliations:** 1Research Group Healthy Ageing, Allied Health Care and Nursing, Hanze University of Applied Sciences, 9714 CA Groningen, The Netherlands; 2FAITH Research, 9714 CA Groningen, The Netherlands; 3Center of Expertise for Lymphovascular Medicine, Nij Smellinghe Hospital, Compagnonsplein 1, 9202 NN Drachten, The Netherlands; 4Experimental Anatomy Research Department, Vrije Universiteit Brussel, 1090 Brussels, Belgium; 5University of Groningen, University Medical Center Groningen, Department of Health Psychology, 9700 RB Groningen, The Netherlands; 6University of Groningen, University Medical Center Groningen, Department of Oral and Maxillofacial Surgery, 9700 RB Groningen, The Netherlands; 7University of Groningen, University Medical Center Groningen, Department of Rehabilitation Medicine, 9700 RB Groningen, The Netherlands

**Keywords:** lipoedema, functioning, ICF, body functions, body structures, activities and participation, environmental factors, personal factors

## Abstract

Lipoedema is a painful non-pitting diffuse “fatty” swelling, usually confined to the legs, that occurs mainly in women. This scoping review aimed to provide an overview of the available research on the functioning of people with lipoedema, according to the International Classification of Functioning, Disability and Health (ICF) framework. Relevant publications and gray literature were retrieved until October 2022. The results sections of each publication were organized using a thematic framework approach. All included studies reported at least one outcome fitting within the domains of body functions and body structures, with most studies focusing on the categories of “sensation of pain”, “immunological system functions”, and “weight maintenance functions”. The ICF domains of activities and participation and environmental factors were mentioned in a small number of the included studies (17 and 13%, respectively), while the domain of personal factors was studied in half of the included studies. In conclusion, the emphasis of lipoedema research is on its description from a disorder-oriented point of view in the form of body functions and body structures, with a lack of information about the other domains of functioning.

## 1. Introduction

Lipoedema is described in the International Classification of Diseases (ICD) as: “non-pitting diffuse “fatty” swelling, usually confined to the legs, thighs, hips and upper arms” [1]. The swelling of subcutaneous fat is disproportionally distributed over the body. The increase in adipose tissue occurs in three or four phases (depending on the classification applied) mainly related to changes in the skin, with the skin surface becoming increasingly irregular as the phases progress [2]. People with lipoedema experience a sensation of heaviness, pain, and spontaneous bruising in the affected limbs [3]. Although pain is an important symptom of lipoedema, its cause has not yet been fully confirmed [4]. There are multiple hypotheses about the cause of lipoedema-related pain and there is increasing evidence for the role of underlying biological changes, inflammation, and hypersensitivity [5]. Lipoedema was first reported in 1940 by Allen and Hines [6], but the etiopathogenesis of lipoedema still remains unclear. Women are particularly affected by this condition, with problems often arising during puberty, pregnancy, or menopause [2,7]. Lipoedema in men is rarely described in the literature. The men in which lipoedema was described usually had an underlying condition associated with higher estrogen levels or lower testosterone levels [8]. The diagnosis of lipoedema is made on the basis of clinical findings and the exclusion of differential diagnoses [9]. Due to the lack of consistent diagnostic criteria, lipoedema is often misdiagnosed and confused with other diseases, leaving the prevalence of lipoedema unclear [2,10]. For instance, lipoedema is often confused with lymphoedema. The name lipoedema suggests the involvement of oedema, but there is conflicting evidence about its role in lipoedema [2,11]. Other differential diagnoses include, for example, obesity, lipohypertrophy, and lipomatosis dolorosa [12].

People with lipoedema can experience problems with daily functioning, not only due to factors such as pain [13,14] but also the presence of mental health problems (e.g., depressive disorders, eating disorders) [15] and impaired physical capacity [9,16]. To understand an individual’s functioning, the World Health Organization (WHO) has developed the International Classification of Functioning, Disability and Health (ICF) [17]. ICF is based on the biopsychosocial model and explains the functioning of individuals with certain health conditions while describing functioning as an umbrella term for body functions, body structures, and activities and participation [17,18]. The WHO defines functioning as the result of interactions between a person’s state of health and environmental and personal factors [17].

Considering people’s functioning from a broad perspective, such as the ICF, corresponds to a dynamic view of health, also called ‘positive health’. This view, suggested by Huber et al., argues that, instead of disease-focused thinking, a more dynamic view of health should be adopted because of the increasingly aging population, the fact that more and more people are living with chronic conditions, and the resulting rising costs [19]. Within this view, health is seen as people’s ability to adapt and self-manage, given life’s physical, emotional, and social challenges. The focus is less on the presence or absence of disease, and more on resilience, functioning, and participation [20].

Current research on lipoedema gives an incomplete picture of the functioning of people with lipoedema, because it seems to focus mainly on the body functions domain of the ICF and includes, for example, studies on etiology, pathogenesis, and diagnosis [8,21,22,23,24]. Review studies focusing on functioning from a broad perspective such as the ICF are lacking. However, to improve the adaptability, self-management, and, thus, functioning of people with lipoedema, better understanding of the effects of lipoedema on all domains of a person’s functioning is needed.

Therefore, in this scoping review, we aimed to provide an overview of the available research on the functioning of people with lipoedema according to the ICF framework in terms of body functions, body structures, activities and participation, as well as environmental and personal factors.

## 2. Materials and Methods

This scoping review was registered in Open Science Framework on 14 March 2022 [25]. The Preferred Reporting Items for Systematic reviews and Meta-Analyses extension for Scoping Reviews (PRISMA-ScR) checklist was used as reporting guideline [26].

### 2.1. Eligibility Criteria

Publications were eligible for inclusion if data were reported from people diagnosed with lipoedema or if the author(s) specifically described that people were included according to the following criteria: (1) bilateral and symmetrical disproportionate fat distribution; (2) persistent disproportionate fat distribution despite weight-loss or raising of extremities; (3) pain, tenderness, and easily being bruised; (4) and minimal pitting oedema [6]. In addition, the publications had to contain original data on functioning as defined by the ICF in terms of body functions, body structures, activities and participation, and environmental factors [17]. Personal factors are not listed in the ICF, but were included in this scoping review according to the list of personal factors developed by Heerkens et al. [27].

Exclusion criteria were: (1) publications about self-diagnosed people with lipoedema and/or about people diagnosed with lipoedema who were also diagnosed with another painful adipose tissue disorder or with lymphoedema; (2) laboratory studies (e.g., biopsy or genetic testing); (3) publications that did not report baseline or pre-treatment data; (4) guidelines, systematic reviews, meta-analyses, news articles, blogs, letters, editorial articles, online comments, videos, and publications recorded in research registries without results; (5) publications in languages other than Dutch, English, or German; (6) publications without full text available. In the case of including two studies by the same first author, which may have been based on the same population and used the same outcome data, the first author was approached to inquire about this population. In the absence of response or in the case of overlapping populations, the most recent study was included in the scoping review.

### 2.2. Information Sources and Search

Publications were gathered from PubMed, Cinahl, Embase, Cochrane, and Scopus to provide full coverage of the literature. The databases were initially searched for eligible publications on 21 February 2022 and an update was performed on 21 June 2022. After this update, the author received notifications of new publications via online publication alert tools based on an underlying saved search string. Study inclusion ended on 1 October 2022. Tailored search strings were built and reviewed by a librarian (Appendix A: Search strings). No limits or filters were used. The reference lists of the selected publications were searched for eligible publications to identify other relevant publications. In addition, databases of gray literature (Bielefeld Academic Search Engine (BASE), Science.gov, and the Lipoedema Foundation LEGATO Lipoedema Library), clinical trial registries (ClinicalTrials.gov and WHO International Clinical Trials Registry Platform), and conference reports and abstracts (Embase) were searched. Websites of specific organizations related to lipoedema (International Lipoedema Association, International Lymphoedema Framework, the Lipoedema Foundation, Wounds International) were also searched by hand for publications and gray literature.

### 2.3. Selection of Sources of Evidence

Potentially relevant publications from all information sources were exported to Endnote [28]. Following de-duplication, two researchers (L.M.K. and R.D.) independently screened the publications for eligibility on title and abstract using Rayyan, a systematic reviews web app for exploring and filtering searches [29]. The screening process was continued with the full text screening process, performed by two researchers (L.M.K. and L.K.). Conflicting assessments and references were discussed until consensus was reached. If no consensus was reached, a meeting with a third researcher (R.D.) took place. Prior to the screening, the process was tested by the researchers involved, using a series of previously excluded references. Cohen’s kappa was used to determine the inter-rater agreement of assessments in the screening process [30].

### 2.4. Data Charting Process

Data from publications judged to have met the eligibility criteria were charted using a data extraction form in Excel (version 2211). The data was extracted independently by one researcher (L.M.K.).

### 2.5. Data Items

The following data was extracted from the publications: (1) authors, publication year, study location; (2) study population (sample size, age, sex, if available, information on onset and duration of disease, stage of lipoedema, and family history); (3) study methodology; (4) outcome measures with regard to functioning (body functions, body structures, activities and participation, personal and environmental factors); and (5) pre-treatment/baseline data with regard to functioning (in terms of body functions, body structures, activities and participation, personal and environmental factors). As far as possible, group averages were extracted. In studies by the same first author with partially the same outcome measures in which was unclear whether the population included was the same and no adequate information was obtained from the author, the potentially overlapping data from that author’s most recent study were used.

### 2.6. Critical Appraisal of Individual Sources of Evidence

The methodological quality was assessed using critical assessment instruments. Two researchers (L.M.K. and A.H.) separately assessed the quality of the included studies. Conflicting assessments were discussed until consensus was reached. If no consensus was reached, a meeting with a third researcher (R.D.) took place. Prior to the quality assessment, the process was tested by the researchers using a series of previously excluded references. A detailed description of the method applied is given in Appendix A: Detailed information on the method of critical appraisal of individual sources of evidence.

The quantitative publications were assessed using the Effective Public Health Practice Project (EPHPP) instrument [31]. The overall assessment was scored in six areas, i.e., selection bias, study design, confounders, blinding, data collection methods, withdrawals, and drop-outs.

The qualitative publications were assessed using the Critical Appraisal Skills Program (CASP) instrument [32]. The ten questions in this instrument are divided into three sections (A. Are the results valid, B. What are the results, C. Will the results help locally/how valuable is the research). The Authority, Accuracy, Coverage, Objectivity, Date, and Significance (AACODS) checklist was used to assess the included gay literature on trustworthiness and relevance [33]. Cohen’s kappa was used to calculate the agreement between the researchers’ assessments. The critical appraisal was not used in data synthesis or to exclude publications.

### 2.7. Synthesis of Results

The characteristics of the included publications and the outcomes measures related to functioning were presented in a summary of findings table, with the results of gray literature presented separately (Appendix A: Study characteristics and Appendix A: Study characteristics gray literature).

The results sections of each publication, including tables and figures, were organized using a thematic framework approach [34]. In this scoping review, the ICF was used as the thematic framework. The method was guided by five steps. The first step was to read the relevant sections carefully and thereby become familiar with the data. In step two, an initial set of codes was created using the second- and third-level classification of the ICF [17]. The list of personal factors developed by Heerkens et al. was used to establish the first set of codes for personal factors [27]. The domains of body functions, body structures, activities and participation, environmental factors, and personal factors were used as main themes. Then, in the third step, the final set of codes was determined during a meeting with the research team. In the fourth step, the relevant results sections were entered in Excel (version 2211)and coded using the WHO’s online ICF browser [35] and the WHO document “International Classification of Functioning, Disability and Health” [17]. Codes for body functions started with ‘b’, codes for body structures with ‘s’, codes for activities and participation with ‘d’, codes for environmental factors with ‘e’, and codes for personal factors with ‘p’. Data were excluded from the scoping review if the study results at hand could not be coded with an ICF code because they were not included in the ICF. Although age and gender are personal factors, these outcomes were not coded as personal factors in this scoping review. Since age and gender are often a standard part of the population description in studies, coding these outcomes biases the representation of other personal factors in studies of lipoedema. However, the outcomes age and gender were reported in the study characteristics table. Finally, in step five, the results were mapped out and summarized narratively. Each step of the coding process was carried out by L.M.K., and in cases of doubt, questions were formulated and presented to the research team to reach consensus. If consensus was not reached, experts in the specific field were contacted for advice.

## 3. Results

### 3.1. Selection of Sources of Evidence

Searching databases and registries generated 1865 records and identification with other methods generated 1178 records (Figure 1). After removing duplicates, a total of 1221 records were screened. Citation screening did not yield any new reports. After the title and abstract screening, 194 reports were assessed for eligibility against the in- and exclusion criteria in the full text screening phase (Cohen’s kappa = 0.29, fair agreement). After full text screening, 53 studies were included for data extraction in this scoping review (Cohen’s kappa = 0.83, strong level of agreement). Most reports were excluded due to a wrong publication type and failure to report pre-treatment data or original data. Thirty-four reports were excluded because people were not properly diagnosed or screened for the diagnostic criteria.

### 3.2. Characteristics of Sources of Evidence

#### 3.2.1. Study Characteristics

In total, twenty-seven cross-sectional studies [15,16,36,37,38,39,40,41,42,43,44,45,46,47,48,49,50,51,52,53,54,55,56,57,58,59,60], ten cohorts [61,62,63,64,65,66,67,68,69,70], three randomized controlled trials [71,72,73], four controlled clinical trials [74,75,76,77], one clinical trial [78], five case report studies [6,79,80,81,82], and one qualitative study [83] were included. Two studies used a cross-sectional and a cohort design to answer the research questions [84,85]. Two non-peer-reviewed studies were included in this review. One of them is a dissertation [73] and the other is a non-published cross-sectional study [59] (Appendix A: Study characteristics and Appendix A: Study characteristics gray literature). The 53 included studies were performed in 13 different countries. Most of the studies were published between 2008 and 2022, with three outliers published in 1951, 1996, and 2001 [6,37,82]. The total number of participants with lipoedema was 3839 in the peer-reviewed literature and 238 participants in the gray literature. Twenty-two studies had one or more control groups and these included a total of 1467 participants.

#### 3.2.2. Participant Characteristics

The mean age of the participants ranged from 32.0–62.0 years in the published literature and 45.8–47.0 in the gray literature (Appendix A: Study characteristics and Appendix A: Study characteristics gray literature). Age at onset was recorded in nine studies and ranged from 12.0 to 24.0 years, and the majority of lipoedema onset occurred during puberty [6,42,47,57,59,60,68,80,82]. All but two studies included exclusively women in the lipoedema group. The two exceptions included one male participant [6,40]. Lipoedema stage was recorded in 38% of the studies, in which the majority of studies included participants with stage two or higher. In two studies, the majority of participants had stage one lipoedema [72,79]. Seven studies evaluated family history and found a positive family history in 16% to 73% of the study population [6,21,57,59,68,73,82].

### 3.3. Critical Appraisal within Sources of Evidence

All but three studies [69,72,76] evaluated with the EPHPP had a global “weak” rating (Cohen’s kappa = 0.71, substantial agreement) (Appendix A: Methodological quality assessment using the Effective Public Health Practice Project (EPHPP) instrument). The vast majority of studies introduced potential selection bias by recruiting participants from specialized clinics without specifying what percentage of the selected individuals participated. In addition, none of the included studies blinded both the outcome assessor and the study participants. The three studies with a “moderate” assessment had a strong or moderate study design and a strong data collection method. Due to the fact that the majority of the studies had a cross-sectional study design, withdrawals or dropouts were scored as “not applicable”. Using the CASP, the one qualitative study scored “yes” on 7/9 items [83] (Appendix A: Methodological quality assessment using the Critical Appraisal Skills Program (CASP) instrument). The questions “If the research design was appropriate to address the aim of the research” and “has the relationship between the researcher and participants been adequately considered” was scored with “can’t tell”. Since only one study was scored using the CASP and agreement was corrected for chance, the Cohen’s kappa for agreement was zero. Of the two studies scored with the AACODS checklist, no study scored “yes” to all items [59,73] (Cohen’s kappa = 0.58, moderate agreement) (Appendix A: Methodological quality assessment using The Authority, Accuracy, Coverage, Objectivity, Date and Significance (AACODS) checklist). One study scored “peer reviewed” because it was a dissertation; however, in this study the qualifications, experience, and reputation of the author, organization, and editor were unclear [73].

### 3.4. Results of Individual Sources of Evidence

All results from individual sources of evidence are presented in a summary of findings tables, which is presented in Appendix A: Study characteristics and Appendix A: Study characteristics gray literature.

### 3.5. Synthesis of Results

Sixteen one-level ICF categories, 42 two-level ICF codes, and six personal factors codes were used to code the relevant result sections of the included studies (Table 1 and Table 2). Table 1 and Table 2 show the codes used for all domains, together with the absolute frequency of the outcomes used in relation to that specific code, the percentage of the total number of studies in which the code has been identified, and the measuring instruments/questionnaires used to quantify those results.

#### 3.5.1. Body Functions and Body Structures

All but one [37] of the studies reported one or more outcomes that fit within the domain of body functions, with “functions of the digestive, metabolic and endocrine systems” coded the most since “b530 weight maintenance functions” (e.g., BMI, waist-to-height ratio) was coded in 79% of the included studies (Table 1). A total of 24 studies reported on “sensory functions and pain”, with 22 of the studies reporting data on pain and 9 studies reporting on “b270 sensitivity to pressure” [6,15,41,42,45,47,57,59,60,61,62,64,65,66,67,68,71,72,73,75,76,78,79,82]. “Functions of the cardiovascular, haematological, immunological and respiratory functions” was reported in 47% of the included studies, because “b435 Immunological systems functions” was used to code 23 outcomes in 18 studies. Most of these outcomes were parameters related to lymphatic function. Furthermore, in this category “exercise tolerance functions” were reported in eight studies, with four studies reporting on general physical endurance (e.g., six-minute walk distance) [16,40,71,73] and four studies reporting on fatiguability [41,42,47,71]. The category “functions of the skin and related structures” was mentioned in 15 studies [41,42,47,54,57,59,60,62,64,65,66,67,73,74,82], whereby “b820 Repair functions of the skin” was used 15 times to code outcomes related to bruising. The category “mental functions” was used to categorize 21% of the included studies [6,41,47,57,62,65,66,67,71,79,83], with five studies reporting on body image coded with “b180 experience of self and time function” [5,61,64,65,66]. “Neuromusculoskeletal and movement-related functions” was mentioned in ten studies [16,41,42,47,57,62,64,66,67,72], with six studies reporting on “b780 sensations related to muscles and movement functions” (e.g., feeling of tension/heavy legs) [42,57,62,64,66,67], three studies on “b710 Mobility of joint functions” [41,47,72] and three studies on “b730 Muscle power functions” [16,41,47]. Of all coded outcomes in this domain, 45% were presented as a percentage of the study population and 27% were described using the visual analogue scale. For the remaining 29% of the outcomes, 24 different measuring instruments (e.g., six-minute walking distance, SF-36) or methods (e.g., two-dimensional echocardiography, lymphoscintigraphy) were used.

Fifteen studies (28%) reported at least one outcome that fit within the domain “body structures”, with 17% of studies reporting structures related to the cardiovascular (e.g., echocardiographic data on the left ventricle) or immune systems (e.g., lymphatic anatomy parameters) [36,37,39,49,50,51,52,53,55,57,58,64,72,84,85]. Six studies (11%) reported on eight outcomes related to “lymphatic anatomy parameters” using five different methods of measurement [36,37,42,50,55,58].

#### 3.5.2. Activities and Participation

The domain of activities and participation is reflected in 9 of the 53 studies (17%) [40,47,57,62,64,65,71,73,78] (Table 2). Five outcomes in five studies were coded with “d450 walking” or “d455 moving around” and all authors used a visual analogue scale to study the outcomes [57,62,64,65,73]. Employment-related outcomes in category “major life areas” were mentioned in two studies [40,62] as well as “d230 carrying out daily routine” [71,78] and “d920 recreation and leisure” [40,73].

#### 3.5.3. Environmental Factors

Seven studies reported on the domain environmental factors, with six of the studies reporting on “e580 health services, systems and policies” [42,62,64,67,72,73,83] (Table 2). Outcomes coded with this code concerned treatments used by the participants of the included studies for their lipoedema problems. One qualitative study reported on experiences of treatment by society and within the healthcare system [83].

#### 3.5.4. Personal Factors

Twenty-seven of the studies reported at least one outcome that fit within the domain personal factors [6,15,36,39,40,41,43,53,57,59,60,61,62,63,64,65,66,68,69,71,72,76,78,79,82,85,86] (Table 2). The majority of outcomes in this domain were coded with “comorbidities” as it was mentioned in 32% of the studies. The code “quality of life” was used in 12 studies to code outcomes related to quality of life. A total of seven different questionnaires/measurement instruments were used to measure this outcome [40,57,59,63,64,65,66,71,72,76,78,79].

## 4. Discussion

The aim of this scoping review was to provide an overview of the available research on the functioning of people with lipoedema, according to the ICF framework in terms of body functions and body structures, activities and participation, environmental factors, and personal factors. Studies on lipoedema mainly focused on the ICF domains body functions and body structures. All included studies reported at least one outcome fitting within one of these domains, with most studies focusing on the categories “sensation of pain”, “immunological system functions”, and “weight maintenance functions”. The ICF domains activities and participation and environmental factors were mentioned in a small number of the included studies (17 and 13%, respectively), while the domain personal factors was studied in half of the included studies.

Studies in lipoedema, which were mostly observational, primarily focused on well-known characteristics such as pain, heaviness, fat distribution, and the presence or absence of oedema. However, this scoping review shows that very little research has been done on the effects of these symptoms on the level of activity and participation—and vice versa—as well as on how environmental and personal factors affect these. The emphasis on the body functions domain in lipoedema research can be explained by the fact that research into lipoedema is relatively new, making research into why and how it occurs a logical first step. In addition, the emphasis on the body functions domain can be further explained by and is consistent with the WHO definition, i.e., “Health is a state of complete physical, mental and social well-being and not merely the absence of disease or infirmity” [87]. This view of health encourages researchers to look at differences and abnormalities between people with conditions and healthy people, and to find outcomes that may or may not necessarily affect the onset of the condition or the functioning of the people with that condition. This scoping review shows that the shift from disorder-oriented thinking to a vision focused on functioning and participation within lipoedema research is insufficient. An important step in this shift is to consider health of people with lipoedema not as a treatment target in itself, but as a means to people doing what they want to do [20]. It is therefore notable that only 17% of the included studies in this scoping review studied an outcome that fit within the activities and participation domain. Within these studies, only half of the categories within this domain were covered. The studies that discussed the domain activities and participation show that people with lipoedema experience problems with mobility, performing daily routines and work-related matters [57,62,64,65,73]. In this light, our observation is that “exercise tolerance functions” are understudied. It is also notable that the categories “self-care”, “domestic life”, and “interpersonal interactions and relationships” from the activities and participation domain are not reflected in the included studies. The included studies showed that pain is a key problem in lipoedema, with pain manifesting mainly in the legs and expressed in various ways (e.g., pulling or dull pain). Because people with lipoedema often experience pain for longer than six months, it can be referred to as chronic pain [2]. Research on chronic pain showed that chronic pain has significant consequences for people, their families, and their social and professional environments [88].

The research of Duenas et al. also showed that several groups of people with chronic pain were severely limited in performing daily activities and physical activities and participation in social activities, and that people with pain have problems with sick leave, having to change occupations, or even losing their jobs. In this scoping review, only two studies reported on outcomes related to maintaining a job of employment [40,62], and the same goes for outcomes related to sports [40,73]. Given the chronic nature of lipoedema pain, it is likely that this group is also experiencing problems with work-related issues and physical activities. Another striking fact is that a total of only seven studies presented an outcome that fit within the environmental factors domain [42,62,64,67,72,73,83]. Six of these were related to the therapies received by participants and only one study presented outcomes related to support and social attitudes. Regarding these environmental factors, Melander et al. showed that people with lipoedema were fat-shamed and judged by others as lazy and characterless [83], and thus, faced social stigma. It is known that stigma is common among obese individuals, that it affects both physical and psychological health, and that it is a problem for effective interventions [89]. Since the vast majority (76–88%) of people with lipoedema struggle with obesity (body mass index (BMI) >30 kg/m^2^), it is likely that this population also has physical and psychological problems due to stigma [47,90].

Regarding personal factors, another observation that emerged from this study was that only 23% of the studies mentioned quality of life. Lipoedema can be considered a chronic condition and living with a chronic condition is known to interfere with an individual’s life in terms of well-being [2]. Quality of life is an important measure to assess the impact of living with a chronic disease on an individual. In addition, it is an important construct within patient-centered care to evaluate the effect of treatment [91]. Furthermore, the low use of validated measurement tools and the heterogeneity in the use of measurement tools within the different ICF domains is noteworthy. Regarding personal factors, for example, a total of seven different measurement instruments were used to determine 12 outcomes related to quality of life. This is consistent with the findings in the study by Czerwinska et al. [92]. They found, for example, that the three studies that measured quality of life had used three different measurement tools for this purpose. This finding could possibly be explained by the lack of specific guidelines for diagnosis and treatment of lipoedema, simply because much knowledge is still lacking. For reproducibility and comparability of studies, it is important that studies use one validated measurement tool to measure outcomes, such as—in the case of measuring quality of life—the Short-Form 36 [93].

The current review has various strengths. First, this scoping review is the first study on people with lipoedema that considered the functioning of this population in light of a more dynamic and positive view of health. Second, a broad systematic search strategy was performed, in both medical databases and research registries as well as in the grey literature. Including databases, relevant websites, and online publication alert tools allowed a broad spectrum of studies to be included in this review (e.g., cross-sectional and longitudinal studies, randomized controlled trials, and a qualitative study). Third, although this is not a requirement when conducting a scoping review, the studies in this review have been critically assessed for quality using a quality assessment tool for a range of quantitative study types so that the quality of the studies could be properly compared. Fourth, the use of the ICF framework facilitated structuring of all research results, which would otherwise have been a challenge. The ICF provides international standard language that can be understood by various healthcare providers.

This review also has limitations that need to be taken into account. First, this review used fairly strict inclusion and exclusion criteria regarding the population. Participants had to be diagnosed with lipoedema, or the inclusion of participant in the included studies had to be described based on specific criteria. Thus, studies that did not properly describe their sample but included relevant information could have been missed. Second, agreement between the two reviewers in the early stages of the screening process was “fair”, as demonstrated by the rather low kappa scores. The low kappa scores can be explained by a structural disagreement on the inclusion of studies with outcomes on the domain “body structures”. In a consensus meeting, the researchers agreed on the inclusion and exclusion criteria related to these studies. Third, personal factors were not specified in the ICF, which may have introduced some bias regarding coding of the outcomes with respect to personal factors. To overcome this, Heerkens and colleagues’ list of personal factors was used to systematize the inclusion of personal factors [27]. Fourth, the data extraction and coding processes were carried out by one researcher only. The choices in this process may have been influenced by the researcher’s perceptions and interpretations. However, to minimize the chance of any potential bias, all steps during the process were regularly discussed and reviewed by the full research team.

Based on this scoping review, a number of recommendations can be made. This study has taken a first step in mapping the available information on functioning of people with lipoedema and, in the process, has revealed a knowledge gap. Studies focusing on outcomes that fit into the domains of activities and participation, environmental factors, and personal factors are lacking. More qualitative and observational studies are needed that provide insight into the level of participation and functioning of people with lipoedema and how lipoedema symptoms affect participation and functioning. Within these studies, study outcomes are needed that fit within the domains of activities and participation as well as environmental and personal factors. To better guide interprofessional treatment guidelines, more high-quality studies with clearly formulated inclusion and exclusion criteria are needed on the effectiveness of treatment methods that contribute to people’s empowerment and self-management. To gain more knowledge on lipoedema, determining appropriate measurement tools is a priority so that study results are more comparable and also reproducible. In the absence of sufficient scientific knowledge to make choices about appropriate measurement tools, consensus among experts in the field of lipoedema is needed. As a result of this scoping review, we recommend that professionals involved in the care of people with lipoedema should be aware of the broadness of the problems this patient group may face in all ICF domains. Although much research remains to be performed in the areas of activities and participation, environmental factors, and personal factors, we recommend that attention be paid to these domains within the diagnostic and therapeutic processes.

## 5. Conclusions

In conclusion, this scoping review provides an overview of the available research on the functioning of people with lipoedema according to the ICF framework in terms of body functions, body structures, activities and participation, environmental factors, and personal factors. Within the research on lipoedema, the emphasis is on the description of lipoedema-related problems from a disorder-oriented point of view in the form of body functions and body structures, with a lack of information about the other domains of functioning. We recommend a shift from disease-focused thinking to a more dynamic view of health in the research on lipoedema and care of people with lipoedema, taking into account all domains of the ICF.

## Figures and Tables

**Figure 1 ijerph-20-01989-f001:**
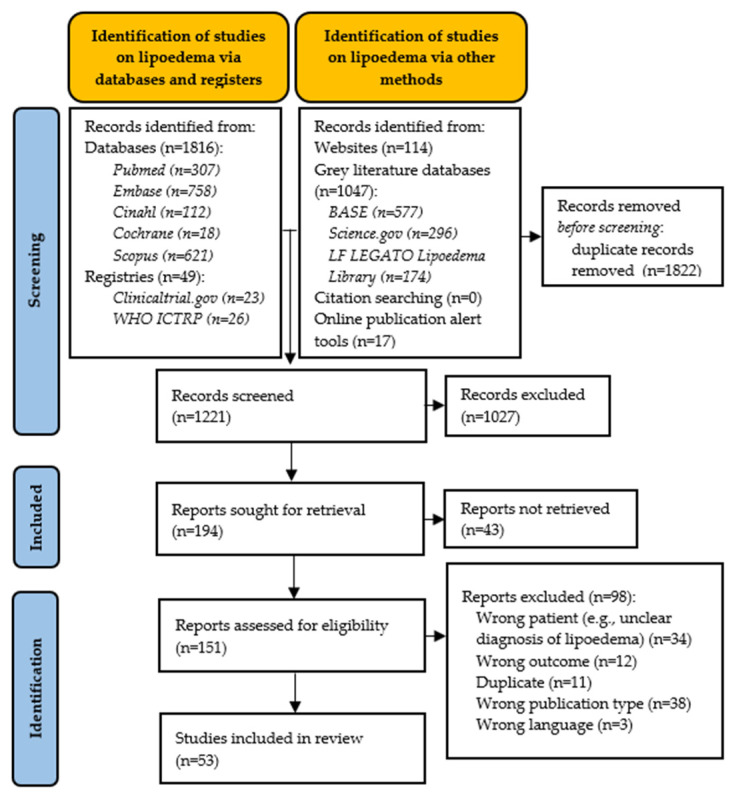
Flowchart search strategy for studies on lipoedema.

**Table 1 ijerph-20-01989-t001:** Frequencies of codes and outcomes used within the theme “body functions” and “body structures”.

	Number of Studies (%)	Number of Outcomes	Outcome Measures/Measurement Instruments
**Body functions**	52 (98.1)	262	
Mental functions	11 (20.8)	13	
b134 Sleep functions	3 (5.7)	3	Sleep Quality Scale (SQS) [79], outcome in % of population [41,47]
b140 Attention functions	2 (3.8)	2	Concentration in % of population [41,47]
b152 Emotional functions	3 (5.6)	3	Beck Depression Inventory [71], Patient Health Questionnaire (PHQ)-9 [57], qualitative data [83]
b180 Experience of self and time functions	5 (9.4)	5	Cosmetic/body image impairment/disturbing body proportions with Visual Analogue Scale (VAS) [62,65,66,67], outcome in % of population [6]
Sensory functions and pain	24 (45.3)	58	
b265 Touch function	2 (3.8)	2	Outcome in % of population [41,47]
b270 Sensory functions related to temperature and other stimuli	9 (17.0)	9	Outcome with VAS [57,62,64,65,66,67], outcome in % of population [63,85], dolorimeter [73]
b280 Sensation of pain	22 (41.5)	47	Outcome in % of population [41,42,45,47,60,72,82], pain with VAS [15,57,61,62,64,65,66,67,68,71,73,75,76,78,79]
Functions of the cardiovascular, hematological, immunological, and respiratory systems	25 (47.2)	50	
b410 Heart functions	4 (7.6)	5	Two-dimensional echocardiography [53,86], Three-dimensional speckle-tracking echocardiography [84,85]
b415 Blood vessel functions	1 (1.9)	1	Venous function in % of population [82]
b420 Blood pressure functions	3 (5.7)	3	Blood pressure measurement [41,53,80]
b430 Hematological system functions	3 (5.7)	7	Outcome in % of population [47], laboratory test results [79,80]
b435 Immunological system functions	18 (34.0)	23	Indocyanine green lymphography/near-infrared fluorescence lymphatic imaging [42,58], non-contrast magnetic resonance lymphography [36], lymphoscintigraphy [46,55,56,80,82], magnetic resonance lymphangiography [50], outcome in % of population [6,41,47,57,59,60,73], outcome with VAS [62,64,73]
b455 Exercise tolerance functions	7 (13.2)	8	Six minute walk test (6MWT) [16,40,71,73], Fatigue Severity Scale (FSS) [71], outcome in % of population [41,42,47]
b460 Sensations associated with cardiovascular and respiratory functions	2 (3.8)	3	Outcome in % of population [41,47]
Functions of the digestive, metabolic, and endocrine systems	44 (83.0)	93	
b525 Defecation functions	2 (3.8)	3	Outcome in % of population [41,47]
b530 Weight maintenance functions	42 (79.3)	58	Bioelectrical impedance analysis (BIA) [77,78], dual-energy X-ray absorptiometry (DXA) [42,73,76], Body mass index [13,14,35,37,38,39,40,41,42,43,44,45,46,47,48,50,52,54,56,57,58,59,61,62,64,65,66,67,68,69,70,71,75,76,77,79,80,83,83,85], body ratio [15,44,59,60,70,71,76,78], outcome in % of population [36,59,82]
b535 Sensations associated with the digestive system	1 (1.9)	3	Outcome in % of population [47]
b540 General metabolic functions	4 (7.6)	6	Indirect calorimetry [76,77], laboratory test result [80], unknown [6]
b545 Water, mineral, and electrolyte balance functions	5 (9.4)	15	Laboratory test result [78,79,80], bioelectrical impedance analysis [43,76], indirect calorimetry [76]
b550 Thermoregulatory functions	1 (1.9)	1	Outcome in % of population [47]
b598 Functions of the digestive, metabolic, and endocrine systems, other specified	3 (5.7)	7	Laboratory test result [78,79,80]
Genitourinary and reproductive function	5 (9.4)	8	
b610 Urinary excretory functions	2 (3.8)	4	Laboratory test result [79,80]
b620 Urination functions	2 (3.8)	3	Outcome in % of population [41,47]
b640 Sexual functions	1 (1.9)	1	Outcome with VAS [66]
Neuromusculoskeletal and movement-related functions	10 (18.9)	19	
b710 Mobility of joint functions	3 (5.7)	3	Beighton score [41,47], outcome in % of population [72]
b730 Muscle power functions	3 (5.7)	3	Outcome in % of population [41,47], hand-held dynamometer [16]
b780 Sensations related to muscles and movement functions	6 (11.3)	13	Outcome in % of population [42], outcome with VAS [57,62,64,66,67]
Functions of the skin and related structures	15 (28.3)	21	
b820 Repair functions of the skin	15 (28.3)	15	Outcome in % of population [41,42,47,59,60,82], outcome with VAS [57,62,64,65,66,67,73], angiostereometry [54,74]
b840 Sensation related to skin	5 (9.4)	5	Outcome in % of population [41,47], outcome with VAS [57,62,64]
b850 Functions of hair	1 (1.9)	1	Outcome in % of population [47]
**Body structures**	15 (28.3)	20	
Structures of the cardiovascular, immunological, and respiratory systems	9 (17.0)	13	
s410 Structure of cardiovascular system	4 (7.6)	5	Two-dimensional echocardiography [53,84,86], three-dimensional speckle-tracking echocardiography [85,86]
s420 Structure of immune system	6 (11.3)	8	Indocyanine green lymphography/near-infrared fluorescence lymphatic imaging [42,58], non-contrast magnetic resonance lymphography [36], fluorescence micro lymphography [37], magnetic resonance lymphangiography [50], lymphoscintigraphy [55]
Skin and related structures	6 (11.3)	7	
s810 Structure of areas of skin	6 (11.3)	7	Ultrasound [39,49,51], outcome in % of population [72], outcome with VAS [57,64]

**Table 2 ijerph-20-01989-t002:** Frequencies of codes and outcomes used within the themes “activities and participation”, “environmental factors”, and “personal factors”.

	Number of Studies (%)	Number of Outcomes	Outcome Measures/Measurement Instruments
**Activities and participation**	9 (17.0)	14	
General tasks and demands	3 (5.7)	4	
d230 Carrying out daily routine	2 (3.8)	3	International Physical Activity Questionnaire (IPAQ) [71], Physical activity level (PAL) [78], steps per day [78].
d240 Handling stress and other psychological demands	1 (1.9)	1	Handling stress in % of population [47]
Mobility	5 (9.4)	5	
d450 Walking	4 (7.6)	4	Impairment in walking with Visual Analogue Scale (VAS) [57,64,65,73]
d455 Moving around	1 (1.9)	1	Impairment in running with VAS [62]
Major life areas	2 (3.8)	2	
d850 Remunerative employment	1 (1.9)	1	Occupational impairment with VAS [62]
d845 Acquiring, keeping, and terminating a job	1 (1.9)	1	Working place (hours/week) in % of population [40]
Community, social, and civic life	2 (3.8)	3	
d920 Recreation and leisure	2 (3.8)	3	Amount of sport activities in % of population [40,73]
**Environmental factors**	7 (13.2)	11	
Support and relationships	1 (1.9)	1	
e355 Health professionals	1 (1.9)	1	Qualitative data [83]
Attitudes	1 (1.9)	1	
e460 Societal attitudes	1 (1.9)	1	Qualitative data [83]
Services, systems, and policies	6 (11.3)	9	
e580 Health services, systems, and policies	6 (11.3)	9	Lipoedema treatments in % of population [42,62,64,72,73], Combined decongestive therapy (CDT) score [67]
**Personal factors**	27 (50.9)	43	
Sociodemographic factors	7 (13.2)	7	
Race	5 (9.4)	5	Race in % of population [6,41,43,59,69]
Education	2 (3.8)	2	Education level in % of population [40,71]
Position in immediate social and physical context	1 (1.9)	1	
Living situation	1 (1.9)	1	Living situation in % of population [40]
Disease-related factors	17 (32.1)	17	
Comorbidities	17 (32.1)	17	Comorbidities in % of population [35,38,39,40,52,56,58,59,60,61,65,67,68,69,81,84,85]
Lifestyle (habits)	2 (3.8)	2	
Smoking habits	2 (3.8)	2	Smoking habits in % of population [40,57]
General ‘mental’ personal factors/ psychological assets	12 (22.6)	16	
Quality of life	12 (22.6)	16	36-Item Short Form Health Survey (SF-36) [40,59,71], Western Ontario and McMaster Universities Osteoarthritis Index (WOMAC) [79], World Health Organization Quality-of-Life Scale (WHOQOL-BREF) [57], The Freiburg Quality of Life Assessment for lymphatic disorders, Short Version (FLQA-lk) [63], Norwegian version of the QoL questionnaire for lymphoedema of the leg [78], Profil der Lebensqualität chronisch Kranker (PLC) [72], outcome with VAS [62,63,64,65,66,67]

## Data Availability

Not applicable.

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
