# Peer review of "Functioning of People with Lipoedema According to All Domains of the International Classification of Functioning, Disability and Health: A Scoping Review"

_ijerph, 2023, doi:10.3390/ijerph20031989_

Round 1

Reviewer 1 Report

 I read with interest the paper titled Functioning of people with lipoedema according to all domains 2 of the International Classification of Functioning, Disability 3 and Health: a scoping review, by Kloosterman et al. This paper is well-written with a lot of work behind the writing and the authors should be commended.

I have minor comments:

  1. Line 43: Consider revising the statement about pain give the recent publication on neuropathic pain in lipedema: PMID 36142221
  2. Lines 52-54: Consider revising the statement that oedema does not play a relevant role in lipoedema when this fluid can be visualized by MR: PMID 35657120; the reference you quote states there clearly is no oedema in lipoedema but PMID 35657120 contradicts this statement
  3. Line 58: Not all people with lipoedema experience mobility issues and eating disorders so please add the word “can” before “experience in the statement “people with lipoedema experience problems”
  4. Line 173: Please explain or make more clear in the text what it means “Will the results help locally”
  5. Please somehow add the word lipoedema to Figure 1.
  6. Line 358-9: You state “It is therefore notable that only 17% of the included studies in this scoping review studied an outcome that fit within the activities and participation domain.” Yet since lipedema research is relatively new, in the last 20 years (Appendix C), do you think perhaps that this early research is focused on getting to know exactly what lipoedema is and what it is not?  Does it not make sense to improve the function of a population that you know, rather than assessing function in a heterogeneous population? It seems that this paper is criticizing papers that are trying to get to know lipoedema. Please clarify.
  7. Lines 395-403: Please state at the end of this paragraph that a single quality of life instrument that can be used in multiple studies will be important to be able to compare studies.  Can you suggest a good QOL questionnaire?

Reviewer 2 Report

Interesting and well carried out meta-analysis about lipedema, a condition difficult to be properly diagnosed and treated nowadays. There is a need for more research about this field.

The methodology and design of the study are correct, several references are included, and the main aim of reviewing the literature on the functioning of people with lipoedema is achieved. There are inclusion and exclusion criteria, the discussion is complete and it includes limitations and strong points of the study. The conclusions are also appropriate.

For all these reasons, I congratulate the authors for their work and I consider this paper suitable for publication in this journal without major changes .

Reviewer 3 Report

Overall, a very important theme, you are discussing here, especially as most of the medical doctors are not familiar with this disease. Unfortunately it´s very hard to read, as most of the text is a description of methods, you used. Maybe this should go in an appendix too. Methods should be much shorter and focused on what you did in general. Introduction is well written but does not fit to the rest of the paper, as it describes mainly lipoedema per se. Only discussion describes the idea of this publication.

Round 2

Reviewer 3 Report

The paper is much better structured now and readable even for people without knowledge about this kind of disease.